# Striatal Cholinergic Signaling in Time and Space

**DOI:** 10.3390/molecules27041202

**Published:** 2022-02-10

**Authors:** Dvyne Nosaka, Jeffery R. Wickens

**Affiliations:** Neurobiology Research Unit, Okinawa Institute of Science and Technology Graduate University, Onna-son 904-0412, Japan; dvyne.nosaka@oist.jp

**Keywords:** acetylcholine, acetylcholinesterase, pause, concentration gradient, synapse, volume transmission

## Abstract

The cholinergic interneurons of the striatum account for a small fraction of all striatal cell types but due to their extensive axonal arborization give the striatum the highest content of acetylcholine of almost any nucleus in the brain. The prevailing theory of striatal cholinergic interneuron signaling is that the numerous varicosities on the axon produce an extrasynaptic, volume-transmitted signal rather than mediating rapid point-to-point synaptic transmission. We review the evidence for this theory and use a mathematical model to integrate the measurements reported in the literature, from which we estimate the temporospatial distribution of acetylcholine after release from a synaptic vesicle and from multiple vesicles during tonic firing and pauses. Our calculations, together with recent data from genetically encoded sensors, indicate that the temporospatial distribution of acetylcholine is both short-range and short-lived, and dominated by diffusion. These considerations suggest that acetylcholine signaling by cholinergic interneurons is consistent with point-to-point transmission within a steep concentration gradient, marked by transient peaks of acetylcholine concentration adjacent to release sites, with potential for faithful transmission of spike timing, both bursts and pauses, to the postsynaptic cell. Release from multiple sites at greater distance contributes to the ambient concentration without interference with the short-range signaling. We indicate several missing pieces of evidence that are needed for a better understanding of the nature of synaptic transmission by the cholinergic interneurons of the striatum.

## 1. Introduction

The cholinergic interneurons of the striatum (CINs) form a local network, specific to the striatum, and distinct from the forebrain cholinergic neurons. Although CINs account for a small fraction of all striatal cell types [1,2,3], they are a rich source of acetylcholine [4], and together with extrinsic afferents [5] give the striatum a higher content of ACh than any other part of the brain, apart from the interpeduncular nucleus [6]. Anatomically, CINs have large somata of approximately 15 to 25 μm in diameter [7], and long, aspiny and infrequently branching dendrites and extensive axonal arborization [8], evident when single neurons are examined (Figure 1A). CINs receive synaptic input from many sources including glutamatergic inputs from thalamostriatal and corticostriatal afferents. The cortical inputs are infrequent [9] and not always observed [10,11] and thalamostriatal inputs play a major role. CINs also receive inputs from dopaminergic axons from the midbrain [12], some but not all classes of GABA interneurons, and collaterals of spiny projection neurons.

The axons of CINs arborize extensively to produce a dense axonal plexus [8]. After branching several times, the axons become extremely fine and difficult to follow using a light microscope. The axons have varicosities containing mitochondria and vesicles [2,7,13], some of which form symmetrical synaptic specializations [2,14,15]. Although these findings established the synaptic nature of cholinergic neurotransmission by CINs in the striatum, Descarries et al. [16,17] on the basis of a systematic electron microscopic analysis of the varicosities [13], reported that fewer than 10% of those varicosities displayed synaptic membrane specializations. Observations such as these, together with studies showing that some ACh receptors are located at non-cholinergic synapses [18,19], suggested the hypothesis of diffuse transmission by ACh [16,17]. Thus, there is evidence for both synaptic and diffuse transmission. 

In other cholinergic systems there have been calls to “move on” from the concept of volume transmission of acetylcholine [20,21], driven in part by a lack of direct evidence for extrasynaptic release, new technology for visualizing temporospatial profiles at millisecond and micrometer resolution [20], and measurement of CIN-mediated activation of G-protein coupled receptors showing fast synaptic transmission [22]. In this review we re-examine quantitative aspects of ACh neurotransmission and, using a simple mathematical model to integrate our findings, estimate how these quantitative details might affect the spatiotemporal dynamics of ACh concentration and thus signaling by CINs in vivo. 

## 2. Firing Patterns of Cholinergic Interneurons and Their Behavioral Correlates

The natural firing patterns of CINs have been extensively studied. In vitro, CINs have depolarized resting membrane potentials [23,24], and show slow, irregular autonomous firing activity at approximately 3–10 Hz [8]. This tonic activity is due in part to the repetitively cycling I_h_ currents generated by hyperpolarization-activated cation nucleotide channels [25,26,27]. These subthreshold oscillations periodically trigger action potentials even in vitro in the absence of excitatory inputs (Figure 1B). In vivo, CIN action potentials appear to be triggered by small excitatory postsynaptic potentials from the cerebral cortex and thalamus [8].

The groundbreaking studies of Kimura and Graybiel [28,29,30] showed that CINs acquire responses to sensory stimuli, such as a solenoid click repeatedly paired with reward (Figure 1C). These responses are acquired gradually and their acquisition is dopamine-dependent [29], but distinct from the responses of dopamine neurons [31,32]. Apicella [33,34] has suggested that CINs have more selective responses, showing changes in firing in response to spatial attributes, such as the location of a stimulus or movement direction [35], the behavioral context [36,37], or the current state of the animal [38]. It is possible, given these responses, that CINs are involved in processing information that is important for flexible switching of responses. Lesion studies also support such involvement in behavioral flexibility. For example, Brown et al. [39] showed that when the thalamic afferents to CINs were inactivated, place learning in a cross maze proceeded normally, but learning was impaired after switching the rewarded side. Similarly, Bradfield et al. [40] found that loss of thalamic afferents to the CINs impaired goal-directed learning after changes in the action–outcome contingency. Aoki et al. [41,42] similarly found that CINs were involved in behavioral flexibility using a set shifting task [43].

**Figure 1 molecules-27-01202-f001:**
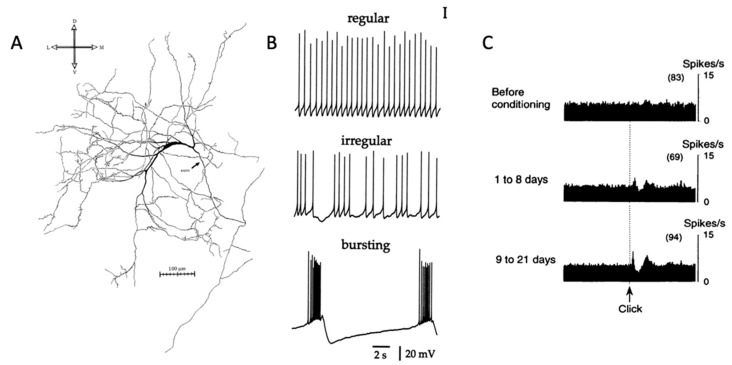
Striatal cholinergic interneuron anatomy and physiology. (**A**) Image of striatal cholinergic interneuron [8]. (**B**) Firing patterns of CIN in brain slice [44]. (**C**) Pause response of presumed CIN to salient stimulus [29]. Permissions requested for use of figures.

How do the responses of CINs to context contribute to behavioral flexibility? In vivo, the firing activity of CINs in behavioral experiments involving both primates and rodents is an inhibitory pause response where the tonic firing activity immediately pauses during learning in response to salient events [45]. As animals are trained to associate cues with reward, more striatal CINs display this pause response, which suggests the CINs’ vital role in learning. The pause response can continue for around 200 milliseconds before returning to the 2–5 Hz firing pattern [34,46,47,48]. At times, there may be a brief excitatory “burst” phase, which may precede or follow the pause. Due to these dynamic firing patterns that involve a pause, with a burst before or after, and then a return to the baseline 2–5 Hz activity, this response to salient cues has been labeled in previous literature as a “three-phase response” [49,50]. There has been considerable interest in the causes of the different phases of the response of CINs and a number of different proposals have been made depending on the context [49,51,52,53,54,55]. In the current review, we focus on ACh signaling, starting from the point at which an ACh-containing vesicle discharges its contents into the extracellular space.

## 3. Striatal Cholinergic Transmission

The nature of cholinergic neurotransmission by CINs is not fully understood. The spatiotemporal distribution of ACh concentration around release sites provides the physicochemical connection between CINs and cholinergic receptors located at downstream locations. Quantifying the dynamics of this concentration distribution requires estimates of the amount of ACh released per spike from the release site, the spatial spread, and the time course of clearance. The key parameters include those related to release, diffusion, and hydrolysis of ACh. Such quantification of the changes in striatal ACh concentration in time and space is fundamental to understanding striatal cholinergic neurotransmission.

### 3.1. Cholinergic Release Sites: Axons, Varicosities, and Synapses

The action of acetylcholine on receptors depends on its physical diffusion from release sites, which is constrained by the tortuosity of the extracellular space, and limited in time and distance by the activity of AChE. The sphere of influence of released acetylcholine extends some distance from the release sites, but how far it extends depends on the resulting concentration distribution from each release site. The degree of overlap of the spheres of influence is also important in determining the steady state or ambient concentration and the specificity of action. To determine the degree of overlap we first need to define the distance between release sites. 

The CINs produce a local axon arborization by repeated division of axons to form a dense feltwork of thin, beaded, ChAT-positive axons and puncta [3]. The axons contain bundles of filaments and some vesicles, and in places expand into varicosities. The varicose expansions of ChAT-positive axons range in diameter from 0.5 to 1.25 µm and contain large electron lucent vesicles and mitochondria [3,15] similar to the varicosities that arise from the neurochemically unidentified large aspiny neurons previously described [14]. Several pieces of evidence, discussed below, indicate that not all varicosities have synaptic specializations, and not all synaptic specializations are on varicosities.

The existence of synaptic specializations on cholinergic varicosities is established, but a wide range of different frequencies of synaptic specializations on varicosities has been reported. Frequent synaptic specializations on axonal enlargements were reported by Groves [56]. Of a total of 440 terminals examined, about 10% contained large rounded or pleomorphic vesicles, and were probably cholinergic because they contained aggregates of synaptic vesicles similar to those seen in ChAT- positive terminals. Most of the terminals examined made synaptic contacts. Similarly, Takagi et al. [14] and Phelps et al. [3] reported that varicosities could often be seen in serial sections to form symmetric type synaptic junctions. In particular, Takagi et al. [14] reported that of 34 varicosities of the presumed cholinergic type (i.e., large round or oval vesicle and medium size mitochondria) arising from large a aspiny neuron, six were synaptic boutons in symmetrical synaptic contact with dendritic shafts or spines, presumably of striatal spiny projection neurons. Wainer et al. [15] reported frequent synaptic specializations that were short (less than 0.2 µm) and predominantly symmetrical, in contact with dendritic shafts and spines. These findings might indicate that a large fraction of cholinergic varicosities of CINs are sites of synaptic specialization. Consistent with this, Groves [56] specifically noted there was no evidence of large numbers of terminals that were not apposed to membranes of postsynaptic targets in the neostriatum.

In contrast to these earlier studies, however, Contant et al. [13] reported that fewer than 10% of ChAT-positive varicosities displayed synaptic membrane specializations. On the other hand, more than 85% of ChAT-positive varicosities contained 10 or more synaptic vesicles. The presence of multiple vesicles in varicosities, the majority of which lack synaptic specializations, suggests that varicosities might be considered to be non-synaptic acetylcholine release sites. If so, then there must be significant non-synaptic release of acetylcholine from ChAT-positive varicosities in the striatum. However, synaptic specializations include not only contacts with the postsynaptic membrane, but also mechanisms for neurotransmitter release, uptake of precursor molecules for acetylcholine synthesis, and regulatory functions such as anchorage for presynaptic receptors. It is not yet clear how these functions are performed by varicosities that lack synaptic specializations, though these mechanisms are not necessarily detectable by microscopy. Another caveat is that membrane specializations typical of cholinergic synapses are small, and may be out of the plane of the section. It is possible that varicosities may appear to be asynaptic because of a low detection rate for cholinergic synapses. 

As a test of logical consistency, the density of ChAT-positive varicosity-located synapses can be compared with an independent estimate of the density of presumed cholinergic synapses. Ingham et al. [57] determined the density of all synapses in the striatum by unbiased stereology and reported that the density of asymmetric synapses in the striatum was 0.925 µm^3^, while the density of symmetric synapses was 0.235 µm^−3^. These figures suggest that 20.3% of all synapses are symmetric. Cholinergic synapses are usually considered to be of the symmetric type. However, there are many types of striatal synapses included in the total number of symmetric synapses, such as GABA, dopamine, and serotonin synapses. Thus, the proportion of symmetric synapses that are cholinergic is not known. Nevertheless, if we assume that 20% of symmetric synapses are cholinergic, it follows that the density of cholinergic synapses is approximately 0.047 µm^−3^. 

This estimated density of cholinergic synapses can be compared with the density of cholinergic varicosities. Contant et al. [13] reported that ChAT-positive varicosities accounted for 19.4% of all neostriatal axon terminals examined. The density of ChAT-positive varicosities, according to Contant et al. [13], is 2.0 × 10^8^ mm^−3^ or *ρ* = 0.20 µm^−3^. If, as reported, fewer than 10% of ChAT-positive varicosities display synaptic membrane specializations [13], the density of synapses on varicosities should be 0.02 µm^−3^. If cholinergic synapses are only located on varicosities, there is a discrepancy with the estimate of cholinergic synaptic density based on total number (0.02 vs. 0.047). This discrepancy could be resolved if the proportion of symmetric synapses that are cholinergic is less than the assumed 20% (e.g., 8.5%). However, Groves [56] suggested that around 10% of all synapses in the striatum were potentially cholinergic, which would indicate that more like 50% of the symmetric synapses are cholinergic. This latter figure may be too high, given that Groves et al. [58] later found that 9% of all synapses were dopaminergic. If these numbers are correct then 19% of all symmetrical synapses are either dopaminergic or cholinergic, which seems too large a proportion given that GABA and other synapses are also included in the 20% of synapses that are symmetrical. Nevertheless, there seems to be no reason to suppose that the proportion of symmetric synapses that are cholinergic is less than the assumed 20%. 

Alternatively the higher density of cholinergic synapses versus cholinergic varicosities with synaptic specializations might indicate that a large fraction (e.g., 60%) of cholinergic synapses are on axons rather than varicosities, as is the case in dopamine axons when carefully examined by serial-section electron microscopy [58]. These considerations suggest that acetylcholine release sites, if exclusively synaptic, might have a density of 0.047 µm^−3^, with a roughly equal distribution between axons and varicosities. This density implies that the mean distance between CIN synapses is about 2.77 µm and the mean nearest neighbor distance is 1.54 µm, assuming random spatial distribution [59]. Spread of acetylcholine from single release sites over greater distances than these would argue against specific, point-to-point signaling, because the clouds of ACh from adjacent release sites would coalesce.

To improve estimates of the density of release sites it is important to determine the proportion of synapses that are cholinergic and the proportion of cholinergic synapses that are on axons vs. varicosities. It is also important to determine what fraction of varicosities release acetylcholine. Even with the benefit of today’s technology, these are technically challenging questions to answer, due to the fundamental limits on the resolving power of light microscopy and the need for serial-section electron microscopy to ensure the plane of section passes through the synapse, as well as difficulties with the neurochemical identification of synapses for electron microscopy, and the high temporal and spatial resolution of acetylcholine concentration distribution in the neighborhood of cholinergic release sites.

### 3.2. Acetylcholine Concentration in Vesicles and Rate of Release

Whether synaptic or non-synaptic, release of acetylcholine involves the fusion of a vesicle with the axonal cell membrane and release of a quantity of neurotransmitter into the surrounding space. Each quantal release event creates a cloud of acetylcholine in the extracellular space, establishing a sphere of influence. In addition to the distance between release sites, the dimensions of this sphere of influence are needed to understand how much overlap between release sites is possible. The number of molecules released per event is an important determinant of the dimensions of the sphere of influence of a given release site. 

Direct measurements of the number of acetylcholine molecules per synaptic vesicle are not available for striatal CINs, but for other central nervous system regions there are estimates of the average number of acetylcholine molecules per vesicle based on biochemical determinations. Vesicles isolated from guinea pig cerebral cortices were estimated to contain about 2.0 × 10^3^ molecules/vesicle [60]. In bovine superior cervical ganglia, an ACh content of 1.63 × 10^3^ molecules/vesicle was estimated [61]. The vesicles in CIN varicosities have been described as follows: “a characteristic type of large round or oval electron-lucent vesicle” [14]; “round vesicles which were generally fairly large but varied in size” [15]; or “pleomorphic synaptic vesicles” [3]. In comparison, the isolated vesicles described by Whittaker [62] measured 469 ± 110 Angstroms, and were morphologically similar to the vesicles in varicosities of CINs. Thus, the best estimate we can make of the ACh content of a single vesicle of a CIN at present is rather tenuous, but a number of 1–2 × 10^3^ molecules/vesicle is plausible.

Another way to estimate the amount of ACh per vesicle is to divide the overall rate of release of ACh by the number of release sites, taking into account the rate of action potential firing and failure rate, to obtain the amount released per spike. Release of ACh in free moving rats has been measured by microdialysis under different conditions [63,64,65]. However, converting the outflow measured in the dialysate into an estimate of the concentration in the extracellular space requires careful consideration of a number of factors that influence this relation. 

The fraction of the extracellular concentration of analyte that is recovered in the perfusate depends on physical properties of the dialysis probe and the rate of removal of the analyte from inside the probe compared to the rate of replacement at the probe membrane surface [66]. In practice, the in vitro recovery fraction, R, is usually determined for a probe before or after it is inserted, and this fraction is then used to make allowance for incomplete recovery when estimating extracellular concentration.

To complicate matters, the recovery fraction is usually measured in solution, in vitro, in the absence of the membranous components that are present in the tissue. The presence of membranous components of the tissue contacting the dialysis membrane, however, has strong effects on the measurement because it blocks contact of the extracellular fluid with the dialysis probe. Thus, how much of the extracellular fluid contacts the dialysis probe depends on the ratio of cellular to extracellular space, and this affects the transport of analyte from the tissue to the probe. In addition, recovery of substances from the tissue depends on the extent to which the convoluted geometry of the tissue increases the apparent diffusion constant by increasing the path length [67]. 

The effects of these factors are not fully known, but can be estimated. Measurements of the interstitial volume fraction α and the tortuosity factor λ have been made using compounds of similar molecular weight to acetylcholine. For example, Nicholson and Phillips [68] found an interstitial volume fraction of α = 0.2 and tortuosity factor of λ = 1.6 using tetramethylammonium. Using such data, theoretical studies (reviewed by Benveniste and Huttemeier [67]) have led to an equation that approximates the relationship between the concentration measured in the dialysate (*C_o_*) and the concentration in the interstitial space (*C_i_*): (1)$$C_{i}=\frac{K\lambda^{2}}{\alpha }\frac{C_{o}}{R}$$

We can apply this equation when estimating the release rate of ACh based on microdialysis data. The CINs are tonically active and this continually releases ACh into the extracellular space, where the ACh is hydrolyzed by AChE. The steady-state concentration at rest is a balance of release and hydrolysis. To measure release, therefore, it is necessary to block hydrolysis. In the presence of AChE blockers, an estimate of release can then be obtained from the rate of increase in concentration (Δ[ACh]/Δ*t*) that occurs when hydrolysis is prevented. 

When hydrolysis of ACh by AChE was blocked using physostigmine (10 µM), Kawashima et al. [69] measured a change in the amount of striatal ACh in the dialysate of awake animals at rest, from 4.98 fM min^−1^ over a 15 min period of perfusion, to 473.2 fM fM min^−1^, over a 5 min period in the presence of the drug, with a perfusion rate of 2 µL/min in both cases. This indicates a concentration increase of Δ[ACh] = 5 × 473.2 fM/2 µL − 15 × 4.9 fM/2 µL = 1146 fM/µL over 15 min. Converting this change over 15 min to a rate of change in concentration per second as measured in the dialysate gives:(2)$$\frac{\Delta C_{o}}{\Delta t}=\frac{1146\times 10^{-15}\;\mathrm{mol}}{900\;s\;\times 10^{-6}\;L}=1.3\times 10^{-9}\;{\mathrm{mol}\;L}^{-1}\;s^{-1}$$

Converting the dialysate measurement to an estimate of the interstitial concentration change, substituting parameters for the interstitial volume fraction of *α* = 0.2 and tortuosity factor of *λ* = 1.6, with *K* = 0.7 [68] and in vitro recovery fraction *R* = 0.22 [69] in Equation (1) gives:(3)$$\frac{\Delta C_{i}}{\Delta t}=\frac{0.7\times {\left(1.6\right)}^{2}}{0.2}\times \frac{1.3\times 10^{-9}}{0.22}\approx 53\times 10^{-9}\;{\mathrm{mol}\;L}^{-1}\;s^{-1}$$

The contribution of each release site per action potential to this increment in concentration can be estimated by dividing by the density of release sites (0.047 µm^−3^ or 0.047 × 10^15^ L^−1^) and the average rate of action potential firing (2 Hz). Multiplying by Avogadro’s constant gives a first estimate of the number of molecules per vesicle, assuming release from one vesicle per release site, and a failure rate of zero. Additional factors to consider in this estimate are the autoregulation of ACh release by muscarinic receptors [70] and inhibition of CIN firing by ACh activation of GABA interneurons [71]. Thus, de Boer et al. [70] showed that the output of acetylcholine was increased to 265% of control values by infusing atropine in the presence of neostigmine, and also increased by 186% in the presence of 10 µM bicuculline [71]. These findings indicate that if the effect of autoinhibition by ACh, and inhibition of CIN firing by activation of GABA neurons were blocked, a roughly five-fold increase in the estimate would be obtained, giving a final value of 946 molecules of acetylcholine per vesicle, in good agreement with the biochemical estimate, as follows:(4)$$n=\frac{6.02\times 10^{23}\times 53\times 10^{-9}}{2\times 0.047\times \;10^{15}}\times 2.65\times 1.86=1673\;\mathrm{molecules}\;\mathrm{of}\;\mathrm{ACh}\;\mathrm{per}\;\mathrm{vesicle}$$

Although both the biochemically based and calculation-based estimates rely on multiple assumptions, the agreement of these two independent estimates gives some confidence in the values obtained. It should be noted, however, that a critical assumption concerns the density of release sites. In the calculation, the density estimated from stereological measures of synapses was used, and a presumed upper bound on the proportion of synapses that are cholinergic was inferred. If, instead, the density of varicosities was used, assuming that all varicosities are release sites, i.e., *ρ* = 0.20 µm^−3^ instead of 0.047 µm^−3^, a much smaller value would be obtained for the number of molecules (*n* = 393). On the other hand, if an estimate of density is calculated based on the finding that only 10% of ChAT-positive varicosities display synaptic membrane specializations [13], i.e., *ρ* = 0.02 µm^−3^ instead of 0.047 µm^−3^, a larger number would be obtained (*n* = 3930). These estimates could be improved with knowledge of the proportion of synapses that are cholinergic, and the proportion of varicosities that are release sites.

### 3.3. Spatial Spread of Acetylcholine: Diffusion and Hydrolysis

In order to estimate the spatial spread of acetylcholine after a release event, it is necessary to take into account diffusion of ACh into the extracellular spaces, and the rate of hydrolysis of ACh by AChE. To gain some insight into the influence of these factors, a conceptual model is helpful, even though there are many unknowns at present. To create such a model, we assume that ACh is released in a punctate fashion from specific locations. The released ACh then diffuses and reacts with AChE. To estimate over what distance acetylcholine spreads after release, we must take into account the rates of release, diffusion, and hydrolysis. As a first approximation, we use the equation developed by Nicholson [72] for behavior of dopamine in the extracellular space, with parameters adjusted for ACh and AChE:(5)$$\frac{\partial C}{\partial t}=\frac{D}{\lambda^{2}}\left(\frac{\partial^{2}C}{\partial r^{2}}+\frac{2\partial C}{r\partial r}\right)-\left(\frac{V_{m}C}{K_{m}+C}\right)$$

This simplified reaction–diffusion model combines Fick’s Law of diffusion with a term for the action of AChE based on Michaelis–Menten kinetics. Here, *C* is the concentration of ACh, *D* is the free diffusion coefficient for ACh, λ is the tortuosity of the tissue as previously defined, *r* is distance from the release site, and *V_m_* and *K_m_* are the standard Michaelis–Menten parameters for AChE. The parameter values used in the model, their units, and the published sources from which they were obtained are shown in Table 1. 

The value of *V_m_* may be calculated from the product of the catalytic constant, or turnover number *K_ca_*_t_ and the concentration of enzyme sites *E_t_*. The turnover number of AChE has been a focus of several studies, in part because it is one of the fastest enzymes known [75]. According to Quinn [81], AChE belongs to a rare class of diffusion-limited enzymes, which means that AChE catalyzes the hydrolysis of ACh so efficiently that the rate limiting step is that of ACh diffusion into the active site. Different values for *K_cat_* have been obtained using different assays, but all give a remarkably large turnover number. For example, values such as 3 × 10^7^ per minute per molecule of enzyme [75], or 7.4 × 10^5^ per minute per active site [76] have been reported; see Rosenberry et al. [77] for a review. 

It should be noted, however, that the effective turnover number in vivo, depends on the substrate concentration. The turnover number is low at low levels of ACh. On the other hand, the fractional rate of hydrolysis is correspondingly greater at these lower concentrations. Wilson and Harrison [76] calculated that quite high concentrations of AChE (micromolar range) would be required to hydrolyze 80% of the ACh present in a millisecond timeframe. While such high concentrations of AChE exist in the neuromuscular junction, in the central nervous system the amount of AChE is much less, and the predominant form is G4-AChE [82]. Quantitative measurement using [^11^C]physostigmine and positron emission tomography indicated concentrations of striatal AChE on the order of 300 nM [78]. This relatively low concentration of AChE in the striatum would be insufficient to have much effect on the immediate time course of ACh concentration after a release event.

Numerical integration of Equation (5) using Mathematica (Wolfram) following a method developed by Huayu Sun [83], and adopting the parameters obtained from the literature (Table 1), provided estimates of the temporospatial dynamics of ACh after release. The discharge of the contents of a vesicle containing 1000 molecules of ACh causes a transient concentration increase in the millimolar range within submicron distances of the release site (Figure 2A). As distances increase, the peak concentration falls rapidly (Figure 2B). This is to be expected as the volume of the extracellular space increases as a cubic function of distance. At a distance of 5 µm the concentration peak just reaches micromolar level about 30 ms after release (Figure 2C).

To have postsynaptic effects, the ACh must be present in an effective concentration at receptors. Receptors for ACh include nicotinic ACh receptors (nAChRs) located presynaptically and muscarinic receptors that may be presynaptic [84,85,86,87,88] or postsynaptic [89,90,91]. We focus here on nicotinic receptors because of their relevance for volume transmission, as presynaptic receptors on non-cholinergic terminals. The two main types of nAChRs in the striatum are the α7 and the α4β2. The receptors with α7 subunits have a high EC_50_ of 117 ± 32 µM [92] whereas the α4β2 have a lower EC_50_ of 1.2 to 93 µM (depending on subunit ratios), [89]. Comparing these values to the concentration distribution shown in Figure 2 indicates where these receptors might be located relative to the release sites. In particular, this suggests that α7 receptors would need to be within sub-micrometer distance from release sites, and thus might have a very specific relationship to one release site at which high µM concentrations could occur. On the other hand, α4β2 would be functionally effective even if located several micrometers from release sites. 

The effect of AChE on the ACh signal shown in Figure 2C is negligible, reflecting the relatively high *K_m_* for AChE (on the order of 100 µM) relative to the ACh concentration changes (on the order of 1 µM). This can be seen by comparing the ACh time course when the reported range of *K_m_* (50–100 µM) is assumed with the ACh time course when *K_m_* is set to an extremely high value (10 mM) in order to model the effects of an AChE blocker. There is essentially no detectable difference in these traces. This was also true for the simulated time and space variations shown in Figure 2A,B (not shown). These findings suggest that AChE plays no significant role in the dynamics of clearance at the micrometer and millisecond spatiotemporal scales relevant to local synaptic transmission. Instead, after release of ACh, diffusion plays the major role in terminating the ACh signal on these timescales of local synaptic signaling.

The lack of effect of AChE on fast, local signaling in the model is consistent with a lack of effect of the AChE inhibitor, eserine, on the time course of excitatory postsynaptic potentials in intracellular recordings from the rat sympathetic ganglion in response to stimulation of preganglionic nerves [82]. A lack of effect of AChE on local synaptic signaling of CINs might also explain the relatively unaffected behavior of proline-rich membrane anchor (PRiMA) knockout mice. In these PRiMA knockout mice, AChE levels in the brain are reduced to 2% of controls [93]. However, they showed no obvious behavioral changes on open field, gait, rotorod, and Morris water maze tests, among others [94]. 

In contrast to its apparent lack of effect on synaptic signaling, AChE appears to play a significant role in shaping the diffusion profile to more distant sites over longer time periods. The estimated concentration profile over distances out to 50 µm from the release site shows sensitivity to the activity of AChE (Figure 2D). If *K_m_* is reduced from 100 µM to 50 µM, the concentration of ACh 1 s after release is reduced. If *K_m_* is increased to 10 mM, the concentration is increased. These effects of AChE activity on remote, delayed concentration profiles are consistent with microdialysis data showing that AChE inhibition causes a ten-fold increase in extracellular levels of ACh compared to baseline conditions [69,73,95]. Similarly, much higher levels of acetylcholine were measured by microdialysis in the perfusate of PRiMA knockout mice (281 ± 167 nM) compared to wild-type mice (1 ± 0.5 nM) [94]. These extracellular measures, however, do not reveal the dynamics of release and clearance at the micrometer and millisecond spatiotemporal scales relevant to synaptic transmission.

### 3.4. Spatial Selectivity of ACh Signaling 

Striatal CINs have an extensive axonal arborization with numerous release sites. The degree of spatial specificity in this context is determined by the ratio of local release effects to the effects of release at other more or less adjacent sites. We can calculate the effects of release from multiple adjacent sites by summing up the amount of ACh released from all sites within diffusion distance of a reference site. If there is a significant difference between a local signal and the combined effect of many distant signals, point-to-point signaling is possible. Conversely, if the signal at the reference release site is indistinct relative to the sum of the signals from distant sites, signaling cannot be spatially selective. 

Although we do not know the density of release sites exactly, our review of the literature suggests a density around 0.040 µm^−3^. Using this estimate, we estimated the spatiotemporal concentration distribution adjacent to a release site that was due to ACh release from that adjacent release site, and compared it with the spatiotemporal concentration distribution at the same location due to ACh release from many distant release sites. This provides a comparison of the local signal with the combined effect of many distant signals. Figure 3 shows our simulation of release events within a 50 µm cube of striatal neuropil in which 5000 release sites have been placed at random to give an average density of 0.040 µm^−3^. For visualization of ACh concentration, a 2D observation plane is placed inside this volume, on which the ACh concentration can be plotted (Figure 3A,B). A single release site placed 0.2 µm above the center of the observation plane provides the reference release site (Figure 3A). The multiple release sites are shown in Figure 3B. The release sites are colored according to which axon they are connected to.

To simulate neural activity, the 5000 release sites are functionally connected to 10 axons, which are activated by tonic activity at 10 Hz. Figure 3(C1) shows the firing times of the axons and Figure 3(C2) shows the corresponding times of the release events, each dot representing a release event at one of the release sites in Figure 3B. Figure 3(D1–D3) show the predicted distribution of ACh concentration at three time points after release of ACh from the single release site. The additional effect of all release events on the ACh concentration at the observation plane after 1.0 s of activity is shown in Figure 3(E1–E3). Comparison of Figure 3D,E shows the degree of specific signaling predicted by the model. The single release site produces a clearly distinct signal even in the presence of ongoing activity at numerous other release sites in the vicinity. Even though the other release sites raise the ambient ACh concentration considerably, the signal from the reference release site stands out. The concentration from all other release sites is less than the peak concentration from the reference site. This shows that given the assumptions of the model, the signal from a release site is spatially specific. 

To understand how the signal from a release site can be spatially specific in the context of multiple active release sites, two opposing factors should be considered. On the negative side, release sites at a greater distance contribute correspondingly less concentration as shown in the simulations of a single release site (Figure 2D) and plotted in Figure 3(F1). On the positive side, at an increasing distance from the reference site there is an increasing number of release sites. The number of release sites within a given distance, *r*, can be estimated based on the density of release sites multiplied by the volume of a sphere with radius *r*. With each increment in *r*, a “shell” containing additional release sites can be imagined. The incremental number of additional sites as a function of distance is plotted in Figure 3(F2). The concentration at the reference site is the product of these two effects. In other words, although the number of sources increases with the increasing distance, they contribute less ACh because of the diffusion distance. We graph the product of these effects in Figure 3(F3). This shows that synchronous activation of all release sites within a certain distance of the reference release site causes an increase of a few tens of micromoles, which is far below the local concentration produced by the reference release site. This concentration is very much higher than would be expected during physiological activity, but is similar to what might be evoked by electrical field stimulation or optogenetic activation of ACh terminals. This rough calculation explains how local signaling can stand out above the cacophony of signals from surrounding release events.

### 3.5. Temporal Resolution of ACh Signaling 

The dynamics of ACh release and clearance have implications for temporal aspects of ACh signaling as well as for spatial aspects. The most distinctive temporal activity pattern of CIN firing in awake animals is a pause in firing, which may be flanked by bursts, sometimes causing a burst-pause-burst sequence. Pauses with a duration in the range of 200–300 ms occur in response to cues that have been repeatedly paired with rewards [31,33,96]. Longer pauses in the spontaneous firing activity of CIN of up to a second have also been reported [90]. At present, it is not known how well changes in ACh concentration track changes in firing rate.

If ACh concentration does track the firing rate of CINs then we would expect induced pauses in CINs to have physiological effects on the spiny projection neurons that are postsynaptic to them. Using optogenetics to silence CINs for brief periods while recording from spiny projection neurons, English et al. [85] found that a pause of 200 ms duration had no immediate effect on the firing rates of spiny projection neurons. However, immediately after the pause a decrease in spiny project neuron firing was observed in association with the rebound bursts of CIN activity that often follow a pause. This CIN burst-related inhibitory effect on spiny projection neurons was caused by the activation of GABAergic interneurons that were excited by the ACh increase associated with the burst, presumably via nicotinic receptors. This finding suggested that there was no direct effect of the pause itself, possibly indicating that ACh was not cleared in the timeframe examined. Apparently in contrast to this earlier work, Zucca et al. [91] found that a 1 s induced pause had an inhibitory effect on spiny projection neurons. They attributed this to removal of cholinergic activation of M1 muscarinic receptors on spiny projection neurons. However, the inhibitory effect only appeared about 400 msec after the onset of the pause. While not contradicting each other, taken together these two results suggest that both a pause and a burst cause inhibition, but via different mechanisms. We consider how this can occur below.

The timeframe within and degree to which pauses in CIN activity influence the concentration of ACh are currently unknown. At the time of writing, we are not aware of empirical studies correlating pauses in the firing activity of CINs with changes in ACh concentration. To gain some insight based on our simple model, we simulated the effects of CIN firing patterns on the time course of ACh concentration during either tonic firing or during irregular firing interrupted by a pause (Figure 4) using a set of observation points within a cube containing 5000 ACh release sites, as before. We found that regular synchronous firing of CINs at 5 Hz (Figure 4(A1)) caused a sawtooth pattern of concentration fluctuations on top of a slowly changing envelope at the observation points (Figure 4(A2)). Underneath these fluctuations, at the start of the firing there was a gradual increase in concentration toward a steady state, and, when firing ceased, there was a gradual decrease, over seconds. In contrast, at an observation point close to a release site, rapidly rising and falling concentration transients were observed. These spikes in ACh concentration occurred in synchrony with each CIN action potential (Figure 4(A3)). The concentration of ACh at the peak of the transients was at least an order of magnitude larger than the slow increase in the ambient concentration envelope measured at other observation points. 

To estimate the effect of a pause on ACh concentration, asynchronous and irregular CIN firing activity was simulated. To simulate a pause, the CIN firing activity was interrupted for 1 s (Figure 4(B1)). The pause was started 6 s after the onset of activity, to allow time for the ACh concentration to reach a steady state. During the pause the concentration of ACh fell, but it took a few hundred milliseconds before the decrease was more obvious than the ongoing fluctuations in concentration (Figure 4(B2)). This suggests that the clearance of ACh is on a timescale of hundreds of milliseconds, limiting the temporal resolution of signaling when receptors are more than a few micrometers distant from release sites. In contrast, close to release sites, receptors would be exposed to fast, large amplitude transients in ACh concentration (Figure 4(B3)).

### 3.6. Limitations and Tests of the Model

A model, as used in the present paper, is a hypothesis, not a proof. Expressing a hypothesis as a model serves to integrate multiple measurements made using a wide variety of techniques into a single formulation. Although highly simplified, it brings different pieces of evidence into juxtaposition. For example, our estimate of the density of release sites, whether from varicosities without synaptic specializations or from classical synaptic junctions, has a reciprocal relationship to estimations of the quantity of acetylcholine stored in a vesicle and discharged by a release event. Given a certain rate of ACh release, estimated from microdialysis, the sum of releases from all individual release sites should add up to the same total whether the density of release sites is 0.2 or 0.04 µm^−3^. Thus, a higher estimate of the density of release sites implies a lower amount of release per site. It is easy to overlook these constraints without a quantitative model.

A model that is defined mathematically has an unambiguous nature, which means that it is more easily testable than a verbally defined model. When we say “volume transmission” or “fast synaptic transmission” it is not always clear what we mean, and it is therefore difficult to design definitive experimental tests of the hypothesis. On the other hand, when we commit to a specific equation and parameter values there is no ambiguity and predictions can be generated that are, in principle, falsifiable. For example, we can make some specific and arguably surprising predictions from the simulation results presented in Figure 2C,D. They predict that blocking the actions of AChE will have no effect on ACh concentration at a distance of 5 µm from the release site in a timeframe of a few hundred milliseconds after a release event (Figure 2C) but that blocking AChE will increase and prolong the ACh signal at greater distances (Figure 2D). An experimental test of this prediction is feasible using genetically encoded sensors expressed at either presynaptic or postsynaptic sites. 

A further benefit of modeling is that definitions can be less granular than verbal descriptions. The concepts of volume transmission and fast synaptic transmission have a dichotomous quality that can lead to polarized viewpoints. As noted by Disney and Higley [97], “existing views often posit a series of false dichotomies (e.g., fast vs. slow, phasic vs. tonic, synaptic vs. non-synaptic)”. The use of a model, even one as simplified as the present one, allows a more nuanced description in terms of concentration gradients of ACh on different spatial and temporal scales. For example, the ACh dynamics depicted in Figure 3 show that large, fast ACh transients tracking spike firing occur close to release sites, whereas slower concentration changes occur at a distance. This analysis suggests a possible explanation for how a CIN can cause inhibition of spiny projection neurons by both a pause and a burst [85,91]. If M1 muscarinic receptors are located more than a few micrometers from release sites, they could be activated by the ambient level of ACh produced by activity at numerous distant release sites, and inactivated when ACh levels fall during a pause, causing a reduction in excitation. Conversely, if nicotinic receptors, such as α7 types [89] expressed on GABA interneurons [98] were located within micrometers of ACh release sites they could be activated by a rebound burst, and cause inhibition. Thus, both a pause and a burst could cause reduced activity of spiny projection neurons.

Conceptual possibilities notwithstanding, a model is only as good as its assumptions. Key assumptions in the current model are as follows: the density of release sites; the quantity of ACh released per spike; the concentration and activity of AChE; and constants such as the diffusion constant for ACh and the tortuosity factor for the striatal neuropil. The derivation of these values and their uncertainty have been discussed in the foregoing review of the literature. In addition to these explicit assumptions, however, there a number of implicit assumptions concerning the enzyme kinetics and spatial distribution of AChE, the firing rate and patterns of activity of CINs, the failure rate of the release sites, and the degree of overlap of axonal territories of different CINs. The equation for the rate of hydrolysis of ACh by AChE is a simplification of the actions of AChE, which has multiple catalytic sites and a complicated reaction mechanism [81]. The model assumes homogeneous distribution of AChE, which may not be the case, and, if AChE is highly concentrated around release sites, it may play a greater role than expected in shaping the ACh signal. The model also does not take into account the uneven distribution of AChE between the patch and the matrix compartments of the striatum [99]. Nevertheless, the actions of AChE in the model are grounded in experimentally determined binding and rate constants, as well as measures of it concentration in the striatum. The model also assumes that ACh synapses are randomly distributed in space, which may not be the case. A firing rate of 5 Hz is assumed, and while this is consistent with published reports, we do not know the failure rate of release events or the degree to which the axonal arborization of CINs overlaps. We also do not know the degree of overlap of axonal territories of CINs. How many different CINs innervate the volume has implications for the degree of synchrony among release events. Finally, the model does not include the effects of autoreceptors on CIN terminals that modulate ACh release, or the effects of GABA interneurons that are activated by ACh [89,98]. Further work is needed to understand the effect of these additional factors on the model’s behavior.

The advent of genetically encoded biosensors for acetylcholine [95,100] has provided a way to examine ACh signaling with high temporal (msec) and spatial (µm) resolution, allowing tests of this model. To date, there has been no extensive study of the temporospatial profile of ACh signaling by CINs. However, some studies have reported on signaling by ACh afferents to stellate neurons in the medial entorhinal cortex in mice [20,100] and other species [95,99]. Of particular relevance, Zhu et al. [20] examined fluorescence changes on the surface of stellate cells at the micron scale, using the genetically encoded sensors for acetylcholine, iAChSnFR and GACh. Electrical stimulation elicited ACh release at multiple, resolvable locations. Although release sites were often clustered together, individual release sites were sometimes seen. The spatial spread of ACh after release could be described by a single exponential with a length constant of ~1.0 μm. Thus, the first findings produced using these new sensors appear consistent with a highly localized distribution of released ACh.

## 4. Conclusions

Despite the many hard-won facts concerning cholinergic signaling by CINs, considerable uncertainty remains. Although we have some evidence concerning the fraction of varicosities that are cholinergic, we do not know for sure whether these are release sites. On the other hand, although we have stereological counts of symmetrical synapses, we do not know what fraction of symmetrical synapses are cholinergic. The best measurements of acetylcholine concentration are currently accomplished using microdialysis, which, although quantitative, has low spatial and temporal resolution. Moreover, translation of dialysate concentrations to extracellular levels requires many assumptions. While the properties of AChE are well studied, we do not have data about the localization and concentration on the fine (µm) scale in striatal tissue. Concerning release events, it is difficult to be sure of the average firing rate of CINs over the entire population, and we do not know the failure rate. The advent of genetically encoded sensors provides the spatial and temporal resolutions required, but these sensors currently do not provide absolute, quantitative measurements, and possibly do not report low nanomolar concentration changes distant from release sites.

Since the seminal work of Descarries et al. [16,17], the concept that ACh acts as a volume-transmitted signal rather than as a rapid point-to-point neurotransmitter has predominated. In this view, ACh is released from varicosities, diffuses, and exerts its effects on multiple cells in a relatively large volume. Considerable indirect evidence supported this hypothesis, including the apparent absence of synaptic specializations on cholinergic varicosities, and the presence of high-affinity receptors distant from cholinergic release sites. However, direct visualization of asynaptic ACh release from varicosities is lacking, and our calculations and recent data from genetically encoded sensors indicate that ACh diffusion is both short-range and short-lived. These considerations suggest that ACh signaling by CIN might be considered a combination of local, transient peaks of ACh concentration adjacent to release sites mediating faithful, time-resolved transmission of spike timing to the postsynaptic cell [22,96] with more distant pooling of ACh to create an ambient level, as proposed by Descarries et al. [16,17].

Whether or not volume transmission is involved, however, it is clear that individual CINs innervate a very large volume of the striatum. Individual CINs thus may control hundreds of thousands of release sites. The potential exists, therefore, for this anatomy to modulate large networks of many thousands of striatal neurons. If this is by local, spatially selective action with steep concentration gradients, then cholinergic neuromodulation can occur in a specific and rapid manner. Resolving this question will be an important step toward elucidating the information-processing operations of striatal CINs.

## Figures and Tables

**Figure 2 molecules-27-01202-f002:**
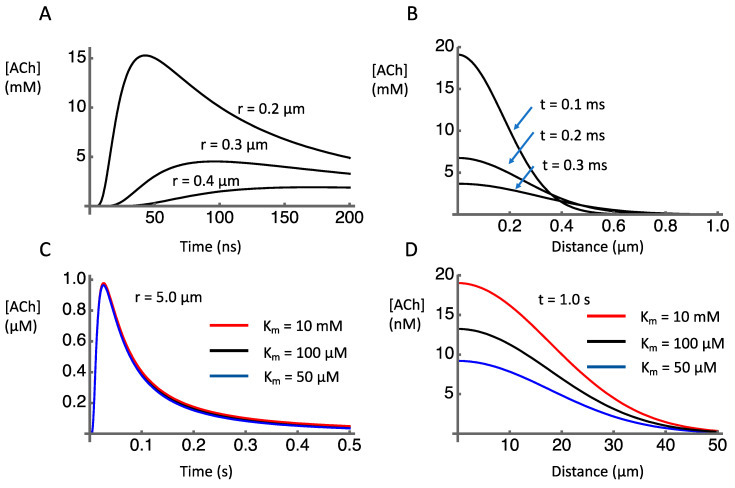
Simulated temporal and spatial concentration profiles after release of 1000 molecules of acetylcholine into the extracellular space. (**A**) Time course of [ACh] at indicated distances from release site. (**B**) [ACh] as a function of distance from release site at indicated time points. (**C**) Time course of [ACh] at 5 µm distance from release site, under conditions of AChE inhibition (red line) or normal (black line) and enhanced (blue line) AChE activity. (**D**) [ACh] as a function of distance from release site 1 s after release, under conditions of AChE inhibition (red line) or normal (black line) and enhanced (blue line) AChE activity.

**Figure 3 molecules-27-01202-f003:**
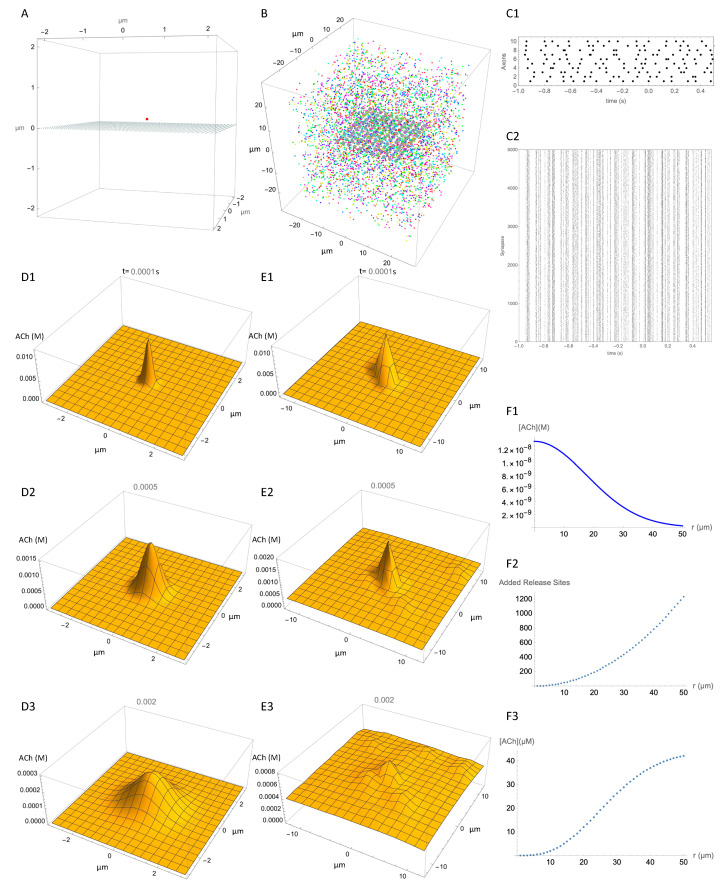
Spatial resolution of ACh signal. (**A**) Reference release site (red dot) positioned 0.2 µm above an observation plane at z = 0, to show spatial spread from a single release site. (**B**) Multiple release sites (typically 5000) distributed within a 50 × 50 × 50 µm cube, with an observation plane at z = 0. (**C1**) Action potential firing times (black dots) for 10 axons. (**C2**) Unsorted release event times (small dots) for each of the 5000 release sites (“Synapses” for convenience of description) over 1.5 s of simulation. (**D**) Still images of the successive temporospatial changes in concentration caused by a release event at the reference release site at time t = 0 s and (**D1**) 0.0001 s, (**D2**) 0.0005 s, (**D3**), 0.002 s. Note different scales for concentration at each time point. (**E**) Same as D but with release events at all 5000 release sites in the block in addition to the reference activity. Timing of release events is as shown in (**C**). (**E1**), 0.0001 s, (**E2**) 0.0005 s, (**E3**), 0.002 s. (**F1**–**F3**) “back of envelope” type calculation of contribution of all release sites to concentration at reference site. See text for explanation.

**Figure 4 molecules-27-01202-f004:**
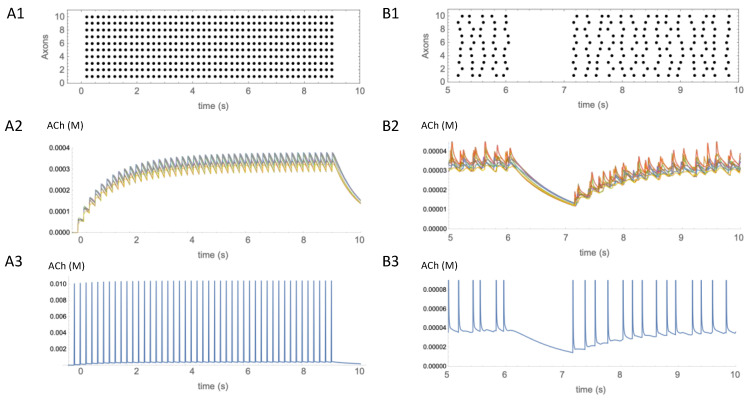
Temporal aspects of ACh signaling. (**A1**) Regular synchronous activation at 5 Hz of 10 axons activated 5000 release sites in the same 50 × 50 × 50 µm volume as shown in Figure 3. Activity started at t = 0 s and continued for 9 s. (**A2**) Estimated concentration of ACh produced by the activity shown in (**A1**). Eight traces showing time course of ACh concentration changes from different observation points in the volume are overlaid. (**A3**) Single trace from an observation point adjacent to a release site, showing the transient increases in ACh at that location. (**B1**) Irregular asynchronous activity at 5 Hz was paused from 6–7s. (**B2**) Eight traces are overlaid showing time course of ACh concentration changes from different observation points in the volume. (**B3**) Single trace from the observation point adjacent to a release site, showing the transient increases in ACh at that location. Transients have the same amplitude as in (**A3**), but have been truncated so that low concentration changes can also be seen.

**Table 1 molecules-27-01202-t001:** Parameter values.

Parameter	Symbol	Value	Units	Sources
Diffusion coefficient	*d*	4.0 × 10^−10^	m^2^ s^−1^	[73]
Tortuosity	*λ*	1.6		[74]
Volume fraction	*α*	0.2		[68]
Molecules/vesicle	*M*	1000		See text
Turnover number	*K_cat_*	1.23 × 10^4^	s^−1^	[75,76,77]
AChE concentration	*E_t_*	300	nM	[78]
Rate constant	*V_m_*	36.9	µM s^−1^	Calculated (*E_t_* × *K_cat_*)
Binding constant	*K_m_*	100	µM	[79,80]

Estimates of the value of *K_m_* for AChE obtained from the literature vary from 0.5 × 10^−4^ to 1.4 × 10^−3^ M [79,80]. These values are high compared to the concentration of acetylcholine, which although not known with confidence, is surely in or below the low micromolar range based on in vivo microdialysis measurements.

## Data Availability

Not applicable.

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
