# Peer review of "Striatal Cholinergic Signaling in Time and Space"

_molecules, 2022, doi:10.3390/molecules27041202_

Round 1
Reviewer 1 Report
In this review, Nosaka and Wickens comprehensively review the recent advance in cholinergic interneurons signaling. The review focus on the spatiotemporal transmission of Ach in the striatum. Authors wants to demonstrate that the concept of ACh acting as a volume-transmitted signal is no longer relevant. For that purpose they take time to review firing pattern of CINs, striatal cholinergic transmission where they discussed potential number and density of release site, Ach concentration and half-life at the synapses. And they build a nice mathematical model to support their demonstration. Overall, the manuscript is balanced, comprehensive, and critical enough for the scope. There are, however, several issues to address and suggestions to consider:
1- Their demonstration of ACh signaling at the spatial level is not convincing. Because their model is based on assumptions about the number of varicosities associated or not with synapses, localisation and density of AChE... Their conclusion shows that the spatial diffusion of ACh is restricted. But they do not take into account the very large arborization of CIN axons and the density of varicosities. These axons will release Ach even if they are not associated directly with synapses. Therefore, the creation of microcompartments seems difficult in these conditions. Of course that volume transmission is clearly not true but microdomain of synaptic Ach signaling also. It might be more accurate to discussed about concentration gradients near the synapse.
- On the other hand, I think that the temporal aspect of Ach signaling is much more important and has not been much taken into account in the review. The fact that the Ach signal is fast and transient can also make sense for the post-synaptic neurones even if the signal is more spread and not focus at the synapse.
- Finally, authors make a paragraph on the firing rate of CINs. But they never talk about the relation that these firing rates could have on dynamics of spatiotemporal Ach signaling.
Author Response
Response to Referee Comments
We thank the referees for their careful reading of the manuscript and helpful comments. We have responded to all of the comments, as detailed below. In brief, we further developed the model to show the degree of spatial specificity in the context of multiple release sites and temporal aspects and effects of firing patters on ACh concentration, and added a section addressing the limitations of the model. We feel the paper is much improved as a result of the referee input.
Referee 1
Referee Comment
Their demonstration of ACh signaling at the spatial level is not convincing. Because their model is based on assumptions about the number of varicosities associated or not with synapses, localisation and density of AChE... Their conclusion shows that the spatial diffusion of ACh is restricted. But they do not take into account the very large arborization of CIN axons and the density of varicosities. These axons will release Ach even if they are not associated directly with synapses. Therefore, the creation of microcompartments seems difficult in these conditions. Of course that volume transmission is clearly not true but microdomain of synaptic Ach signaling also. It might be more accurate to discussed about concentration gradients near the synapse.
Author Response
We agree that we need to take into account the very large arborization of CIN axons and the density of varicosities. We have added a new section entitled “Spatial selectivity of ACh signalling” in which we take into consideration the density of varicosities and effects of many nearby release events. We agree with the language of “concentration gradients” near release sites and have modified the text where mention of synaptic or volume transmission occurs.
Referee Comment
On the other hand, I think that the temporal aspect of Ach signaling is much more important and has not been much taken into account in the review. The fact that the Ach signal is fast and transient can also make sense for the post-synaptic neurones even if the signal is more spread and not focus at the synapse.
Author Response
We agree that the temporal aspect of ACh signalling is important and have added a section entitled “Temporal resolution of ACh signalling” in which we discuss the temporal aspect of ACh signaling.
Referee Comment
- Finally, authors make a paragraph on the firing rate of CINs. But they never talk about the relation that these firing rates could have on dynamics of spatiotemporal Ach signaling.
Author Response
In the new section we discuss the relation between firing rate changes and the dynamics of spatiotemporal ACh signalling.
Reviewer 2 Report
In this manuscript, the authors review the evidence for the theory concerning striatal cholinergic interneuron signaling telling that numerous varicosities on the axon produce an extrasynaptic, volume-transmitted signal rather than mediating rapid point to point synaptic transmission. They used a mathematical model to integrate the measurements reported in the literature and estimated theoretically the temporospatial distribution of acetylcholine after release from a synaptic vesicle. They concluded that the temporospatial distribution of acetylcholine is both short-range and short-lived, and dominated by diffusion.
It's a well-written manuscript, and I only have minor comments.
- The manuscript would benefit from some discussion on the limitation of the mathematical model used here.
- There is a problem with numbering of Tables and Figures. There are two Figures 1 and no Figure 2. The only Table is numbered as Table 2.
- Please check for typos and correct them carefully, eg. vescicle (line 16)
Author Response
Response to Referee Comments
We thank the referees for their careful reading of the manuscript and helpful comments. We have responded to all of the comments, as detailed below. In brief, we further developed the model to show the degree of spatial specificity in the context of multiple release sites and temporal aspects and effects of firing patters on ACh concentration, and added a section addressing the limitations of the model. We feel the paper is much improved as a result of the referee input.
Referee 2
Referee Comment
The manuscript would benefit from some discussion on the limitation of the mathematical model used here.
Author Response
We agree and have added a section “Strengths, limitations and tests of the model“ concerning the limitations of the model, combined with discussion of strengths and potential tests of the model.
Referee Comment
There is a problem with numbering of Tables and Figures. There are two Figures 1 and no Figure 2. The only Table is numbered as Table 2.
Author Response
Sorry about that. The numbering of Table and Figures has been corrected
Referee Comment
Please check for typos and correct them carefully, eg. vescicle (line 16)
Author Response
Typos have been corrected.